# Interferon-β Intensifies Interleukin-23-Driven Pathogenicity of T Helper Cells in Neuroinflammatory Disease

**DOI:** 10.3390/cells10082139

**Published:** 2021-08-20

**Authors:** Agnieshka Agasing, James L. Quinn, Gaurav Kumar, Robert C. Axtell

**Affiliations:** Department of Arthritis and Clinical Immunology, Oklahoma Medical Research Foundation, Oklahoma City, OK 73104, USA; a.m.agasing@gmail.com (A.A.); jmsqnn89@gmail.com (J.L.Q.); gaurav-kumar@omrf.org (G.K.)

**Keywords:** experimental autoimmune encephalomyelitis, T helper 17, interferon-beta

## Abstract

Interferon (IFN)-β is a popular therapy for multiple sclerosis (MS). However, 25–40% of patients are nonresponsive to this therapy, and it worsens neuromyelitis optica (NMO), another neuroinflammatory disease. We previously identified, in both NMO patients and in mice, that IFN-β treatment had inflammatory effects in T Helper (TH) 17-induced disease through the production of the inflammatory cytokine IL-6. However, other studies have shown that IFN-β inhibits the differentiation and function of TH17 cells. In this manuscript, we identified that IFN-β had differential effects on discrete stages of TH17 development. During early TH17 development, IFN-β inhibits IL-17 production. Conversely, during late TH17 differentiation, IFN-β synergizes with IL-23 to promote a pathogenic T cell that has both TH1 and TH17 characteristics and expresses elevated levels of the potent inflammatory cytokines IL-6 and GM-CSF and the transcription factor BLIMP. Together, these findings help resolve a paradox surrounding IFN-β and TH17-induced disease and illuminate the pathways responsible for the pathophysiology of NMO and MS patients who are IFN-β nonresponders.

## 1. Introduction

IFN-β remains a popular treatment for relapsing-remitting multiple sclerosis (RRMS). However, despite the clinical efficacy of IFN-β, approximately 25–40% of RRMS patients respond poorly to this treatment [1], and it consistently worsens disease in patients with neuromyelitis optica (NMO), a neuroinflammatory disease often misdiagnosed as MS [2,3]. These observations demonstrate that there are limitations to the therapeutic benefits of IFN-β.

IFN-β is a pleiotropic cytokine that has many effects on the immune system. Although the mechanism of action of IFN-β in MS is currently unclear, some have speculated that the efficacy of this therapy is achieved by inhibiting the function of T Helper (TH) 17 cells [4,5,6]. However, biomarker studies of MS and NMO patients, as well as animal experiments, have confounded this theory.

In the MS population, there are clear responders and nonresponders to IFN-β therapy. The differences in response in patients have been associated with therapeutic noncompliance [7], the development of neutralizing antibodies [8], as well as differences in the immunological signatures of patients [9]. We previously identified that the balance of TH1 to TH17 signatures determined the efficacy of IFN-β therapy. High TH1 signatures and high levels of IL-7 before initiation of therapy are associated with a good response to IFN-β [10,11]. Alternatively, elevated TH17 cytokine signatures in a subset of MS patients are associated with nonresponders [9,11,12]. In NMO, a disease with a strong association with TH17 signatures [13], patients have an increased frequency of relapses when treated with IFN-β [2,3,14].

We recently reported, in both NMO patients and in mice, that IFN-β treatment had inflammatory effects in T Helper (TH) 17 induced through the action of the inflammatory cytokine IL-6 [15]. However, other studies have shown that IFN-β inhibits TH17 [4,5,6]. This manuscript now provides data to reconcile the conflicting reports of IFN-β on the differentiation and pathogenic function of TH17 cells in neuro-autoimmune diseases.

## 2. Materials and Methods

### 2.1. Mice

Eight- to ten-week-old female C57BL/6J mice were purchased from Jackson Laboratory, housed in the Oklahoma Medical Research Foundation animal facility and treated in compliance with the Institutional Animal Care and Use Committee (IACUC).

### 2.2. Culturing Encephalitogenic T Cells

Donor C57BL/6 mice were subcutaneously immunized with 150 μg MOG_35–55_ peptide (Genemed Synthesis Inc., San Antonio, TX, USA) emulsified in Complete Freund’s adjuvant (5 mg/mL heat-killed *M. tuberculosis*). This was followed by an intraperitoneal (I.P.) injection of 250 ng of *Bordetella pertussis* toxin (List Biological Laboratories Inc., Campbell, CA, USA) in PBS at day 0 and day 2 post-immunization. Donor mice were sacrificed day 10 post-immunization, and spleens and lymph nodes were harvested and mechanically disrupted to obtain a single-cell suspension. 2.5 × 10^6^ cells/mL were stimulated for three days with MOG_35–55_ (10 µg/mL), IL-23 (10 ng/mL; R & D Systems, Minneapolis, MN, USA) and anti-IFN-γ (10 µg/mL; eBioscience, San Diego, CA, USA) in the presence or absence of IFN-β (100 U/mL; PBL, Novato, CA, USA) in complete RPMI 1640 (Gibco, Waltham, MA, USA). To block IL-6 signaling, cells were cultured with anti-IL-6 (10 μg/mL; BD Biosciences, Franklin Lakes, NJ, USA).

### 2.3. Adoptive Transfer EAE

C57BL/6J recipient mice were I.P. injected with 5 × 10^6^ cells and monitored daily for clinical scores. Paralysis was assessed using the following standard clinical score: (0) healthy, (1) loss of tail tone, (2) partial hind-limb paralysis, (3) complete hind-limb paralysis, (4) forelimb paralysis, and (5) moribund/dead. Transfer EAE mice were sacrificed at day 15, and spinal cords were fixed and sectioned for histological analysis using H & E and Luxol fast blue staining.

### 2.4. Isolation of CNS-Infiltrating Cells

Mice were perfused with PBS, and spinal cords were collected and homogenized through mechanical disruption. CNS homogenates were incubated with DNAse (5 µL/mL; Sigma) and collagenase D (4 mg/mL; Roche) at 37 °C for 1 h, and cells were isolated using a Percoll gradient.

### 2.5. Flow Cytometric Analysis of Cytokine Expression

Cells were stimulated with 50 ng/mL PMA (Sigma-Aldrich, St. Louis, MO, USA), 500 ng/mL ionomycin (Sigma-Aldrich), and monensin (BD Biosciences) for 4 h as described by the manufacturer’s protocol. Cells were then stained with CD4 (eBioscience) or CD19 (eBioscience), fixed and permeabilized with Cytofix/Cytoperm (BD Biosciences) and stained for IL-17 (BioLegend), IFN-γ (BioLegend) and/or IL-6 (eBioscience). Data was collected on LSRII (BD Biosciences) and analyzed using FlowJo software (Tree Star Inc., Ashland, OR, USA).

### 2.6. Analysis of Cytokine Production by ELISA

Culture supernatants were collected for ELISA. IL-17, IL-10, IL-6 and GM-CSF levels were assessed using anti-mouse ELISA kits (eBioscience).

### 2.7. RNA Isolation and Quantitative Real Time RT-PCR

RNA was extracted from cells with an RNAeasy Mini Kit (QIAGEN). cDNA was generated using an iScript cDNA synthesis kit (Bio-Rad, Hercules, CA, USA). qRT-PCR was performed using forward and reverse primers (see Appendix A), cDNA and iQ SYBR Green Supermix (Bio-Rad) on a 7900HT Fast Real-Time PCR System (Applied Biosystems, Waltham, MA, USA). Sample reactions were carried out in triplicate. GAPDH was used as a reference gene, and the relative gene expression analysis was measured using the 2-ΔΔCt method [16].

### 2.8. Intracellular Staining of Phosphorylated STAT Proteins

Cells from the spleen and lymph nodes of MOG-immunized mice were cultured as described above. After 0, 15, 30 and 60 min, cells were fixed with 4% paraformaldehyde for 10 min, centrifuged at 1500 rpm for 5 min and permeabilized on ice with 100% cold methanol for another 10 min. Cells were then washed with PBS and stained with the following antibodies: anti-CD4 (eBioscience), pSTAT1 (BD Biosciences), pSTAT3 (BD Biosciences), pSTAT4 (eBioscience), pSTAT5 (eBioscience) and pSTAT6 (eBioscience) for flow cytometric analysis.

### 2.9. In Vitro TH17 Differentiation

Spleen cells from healthy mice were depleted of CD8^+^ T lymphocytes using CD8a (Ly-2) MicroBeads (Miltenyi Biotec) and cultured in two sequential phases. For TGF-Beta-dependent TH17 differentiation, cells were cultured (5 × 10^6^ cells/mL) with antibodies against CD3 (eBioscience; 10 µg/mL) and CD28 (eBioscience; 0.5 µg/mL) and the cytokines IL-6 (R & D Systems; 20 ng/mL) and TGF-β (R & D Systems; 1 ng/mL) in the presence or absence of IFN-β (100 U/mL) for three days. For IL-23-dependent TH17 differentiation, the cells initially cultured with IL-6 and TGF-β (without IFN-β) were washed and recultured for an additional three days with anti-CD3 (10 µg/mL), anti-CD28 (0.5 µg/mL), IL-23 (10 ng/mL) and anti-IFN-γ (10 µg/mL) with or without IFN-β (100 U/mL).

### 2.10. Statistical Analysis

Data are presented as means ± s.e.m., and statistical significance was determined using Mann–Whitney tests. In the case of three or more data sets, means were compared using the Kruskal–Wallis test with a Dunn’s multiple comparison correction. Differences were considered significant for *p* < 0.05. Statistical analyses were performed using Prism 6 (GraphPad, San Diego, CA, USA).

## 3. Results

### 3.1. IFN-β Potentiates the Pathogenicity of Myelin-Enriched TH17 Cells

Previous studies have shown that IFN-β treatment worsened TH17-induced EAE [11]. To understand how IFN-β increased TH17 disease severity, we assessed the direct effects that IFN-β stimulation had on the encephalitogenic capacity of IL-23-derived TH17 cells. We cultured donor cells with MOG_35–55_ and IL-23 in the presence or absence of IFN-β prior to the transfer into healthy recipient mice. Mice which received IL-23- and IFN-β-stimulated cells showed exacerbated disease scores compared to recipient mice that received cells stimulated with only IL-23 (Figure 1A). In agreement with the clinical scores, the histological analysis of spinal cords indicated that mice receiving IFN-β-stimulated cells had increased CNS-inflammation compared to mice receiving cells stimulated with only IL-23 (Figure 1B).

We next examined the production of IL-17 and IFN-γ in the CNS of recipient mice 15 days following adoptive transfer. In the spinal cords, mice receiving cells stimulated with IFN-β and IL-23 had significantly elevated numbers of IFN-γ^+^IL-17^-^ and IFN-γ^+^IL-17^+^ TH cells compared to mice that received cells stimulated with only IL-23 (Figure 1C–E). The differences in the numbers of IFN-γ^-^IL-17^+^ TH cells were nearly significant between both groups of mice (Figure 1F).

### 3.2. IFN-β Induces Inflammatory Cytokines in Myelin-Enriched TH17 Cells

It has been reported that IFN-β inhibits IL-17 expression [4,5,6]. However, we detected increased numbers of IL-17^+^ TH cells in the CNS of recipient mice that received IFN-β-treated cells. Therefore, we next determined the phenotype of the cells before they were transferred into recipient mice. Strikingly, we observed that CD4^+^ T cells treated with IFN-β and IL-23 had increased IL-17 expression measured by FACS, qRT-PCR and ELISA compared to CD4^+^ T cells treated with only IL-23 (Figure 2A–C). We also assessed the expression of other inflammatory cytokines. We observed that in IL-23-stimulated conditions, IFN-β significantly elevated IL-6, although the amounts were substantially lower than IL-17 (Figure 2D). T cells are not thought of as a major source of IL-6. In these culture conditions, we observed that IFN-β increased IL-6 production in B-cells in a dose-dependent manner (Figure 2E). Recently, it has been reported that GM-CSF is a key cytokine produced by TH cells which promotes neuroinflammation [17]. Here, we show that IFN-β significantly elevates GM-CSF expression in IL-23-stimulated conditions (Figure 2F).

IL-6 is a critical factor for TH17 differentiation and IL-17 expression [18]. Our data demonstrating that IFN-β increased the expression of IL-17 and IL-6 led us to hypothesize that IFN-β indirectly induced IL-17 through IL-6. We tested this hypothesis by inhibiting IL-6 in these cultures with a neutralizing antibody. We observed that blocking IL-6 expression had no effect on IL-17 expression (Figure 2G). However, we did find that IL-6 inhibition significantly reduced GM-CSF expression (Figure 2H). We recently reported that in vivo administration of IFN-β exacerbated paralysis and elevated GM-CSF^+^ TH cells in CNS mice with TH17-EAE [15]. Furthermore, the therapeutic blockade of IL-6 reversed these effects of IFN-β [15]. Our in vitro data now provide further evidence that IFN-β-mediated induction of GM-CSF is partially dependent on elevated IL-6 expression. In summary, these results show that IFN-β, with IL-23, induces a highly inflammatory cell population.

### 3.3. IFN-β Induces Key Transcription Factors for the Differentiation of Pathogenic T Cells

We observed that IFN-β increased inflammatory cytokine production from T-helper cells. Therefore, we determined if transcription factors important for the differentiation of pathogenic T-cell populations were altered by IFN-β. Transcription factors for TH17 and TH1 differentiation have been shown to promote the inflammatory potential of T-cells [19,20,21]. We found that IFN-β significantly upregulated the TH17-lineage transcription factors RORγt and Runx1 (Figure 3A,B), which was consistent with our observation of an increased IL-17 production by IFN-β. We also observed that the TH1-lineage transcription factor Runx3 (Figure 3C), but not T-bet (Figure 3D), was significantly elevated by IFN-β. Interestingly, it has been suggested that cells that have both TH1 and TH17 phenotypes are highly pathogenic. In a recent study, Blimp-1 was identified as a crucial IL-23-induced transcription factor that helped drive TH17-mediated inflammation through the induction of GM-CSF [22]. We observed that Blimp-1 expression was markedly increased in IFN-β- and IL-23-stimulated cells compared to cells stimulated with only IL-23 (Figure 3E). IFN-β signals through the JAK-STAT pathway, primarily through STAT1. However, IFN-β can also activate other STAT transcription factors, including STAT3 [23]. STAT1 signaling is required for TH1 differentiation, and STAT3 is required for TH17 differentiation [24,25]. We found that IFN-β resulted in the intracellular phosphorylation of STAT1 (Figure 3F). We found no induction of STAT3, STAT4, STAT5 or STAT6 phosphorylation by IFN-β (Figure 3F). Our data demonstrate that IFN-β promotes a phenotype that has both TH1 and TH17 characteristics, a phenotype shown to be highly pathogenic [26].

### 3.4. IFN-β Has Differential Effects on TGF-β-Dependent and IL-23-Dependent TH17 Differentiation

The data described above contradict reports showing that IFN-β inhibits TH17 development. In culture, TH17 cells can develop in the presence or absence of TGF-β or IL-23 (Figure 4A). TGF-β, with IL-6, initiates the differentiation of naïve T cells to TH17 cells; however, these TH17 cells are not highly inflammatory [27]. IL-23 is also involved with the differentiation of TH17 cells and is essential in driving the inflammatory functions of these cells [27,28]. We speculated that IFN-β may have different effects on these discrete pathways of TH17 differentiation. First, we assessed the effect of IFN-β on TGF-β-mediated TH17 differentiation. We differentiated resting CD4^+^ T cells with IL-6 and TGF-β and found that IFN-β significantly reduced the expression of IL-17 and GM-CSF in these cultures (Figure 4B,C). Next, we assessed the effects of IFN-β on IL-23-mediated TH17 development. Here we took the initial TH17 cultures (TGF-β and IL-6 without IFN-β) and restimulated the cells with IL-23 in the presence or absence of IFN-β. In this condition, we found that IFN-β significantly increased the expression of IL-17 but had no effect on GM-CSF expression (Figure 4B,C). These data demonstrate that TGF-β and IL-23 are important factors that impact the function of IFN-β on TH17 development.

## 4. Discussion

These new data help to rectify the apparent conflicting theories on how IFN-β affects TH17 function. One predominant theory behind the efficacy of IFN-β is that this therapy reduces disease by inhibiting TH17 differentiation and function [4,5,6]. However, MS and NMO patients with high TH17 signatures and mice with TH17-induced EAE have exacerbated disease when treated with IFN-β [2,3,9,11,13,14,29]. Our new data provide key insights into how IFN-β paradoxically increases TH17 pathology. IFN-β has differential effects on discrete pathways of TH17 development. In accordance with previous reports, IFN-β inhibits IL-17 production; however, this is during TGF-β-dependent TH17 development, which does not produce a pathogenic cell population [27]. In contrast, IFN-β enhances the pathological functions of T cells during TH17 development, which is driven by IL-23 [27,28]. Our experiments define a mechanism where IFN-β with IL-23 induces a highly pathogenic T helper cell population that expresses elevated levels of Blimp1, the TH17-lineage genes RORγt and Runx1, and the TH1-lineage gene Runx3. These pathogenic T cells secrete elevated levels of IL-6 which contribute to the secretion of the inflammatory cytokine GM-CSF.

This study provides key insights into how IFN-β drives pathology in diseases with elevated TH17 signatures. Our observations suggest that IFN-β with IL-23 can alter both the transcriptional and cytokine profiles towards an inflammatory phenotype during TH17-mediated disease. These findings provide further evidence that IFN-β may worsen in certain neuro-autoimmune diseases, such as in patients with NMO.

## Figures and Tables

**Figure 1 cells-10-02139-f001:**
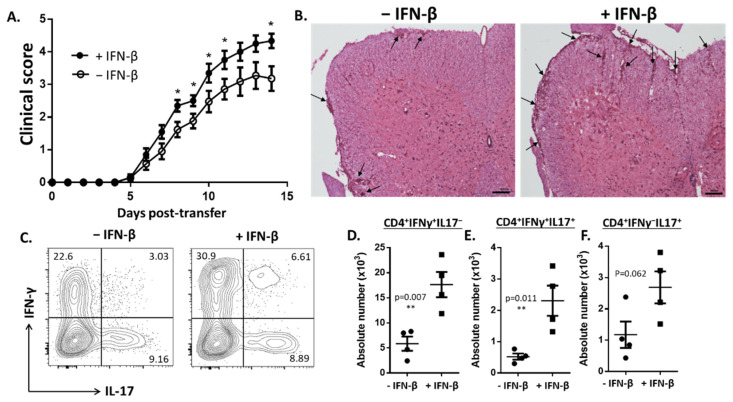
IFN-β increases the pathogenicity of IL-23-stimulated, myelin-enriched TH17 cells. (**A**) Clinical scores of mice with EAE induced by adoptive transfer of splenocytes cultured with MOG_35–55_, IL-23 and anti-IFN-γ in the presence (+) or absence (−) of recombinant mouse IFN-β (*n* = 20–21 mice per group from three independent experiments). (**B**) Histology of spinal cord sections from representative mice from the − IFN-β and + IFN-β groups. Sections were obtained 15 days post-transfer and stained with H & E. Arrows indicate demyelinating lesions in the brain and spinal cord. (**C**) Representative flow cytometry plots of CD4+ T cells expressing IFN-γ and IL-17 in the spinal cords of EAE mice, 15 days post-transfer. Absolute numbers of T helper cells that are (**D**) IFN-γ^+^IL-17^-^, (**E**) IFN-γ^+^IL-17^+^ and (**F**) IFN-γ^-^IL-17^+^ in the spinal cords of EAE mice. Error bars represent means ± s.e.m. Statistical analysis was performed using Mann−Whitney tests. *p* values < 0.05 were considered significant (** indicates *p* ≤ 0.01, * indicates *p* < 0.05).

**Figure 2 cells-10-02139-f002:**
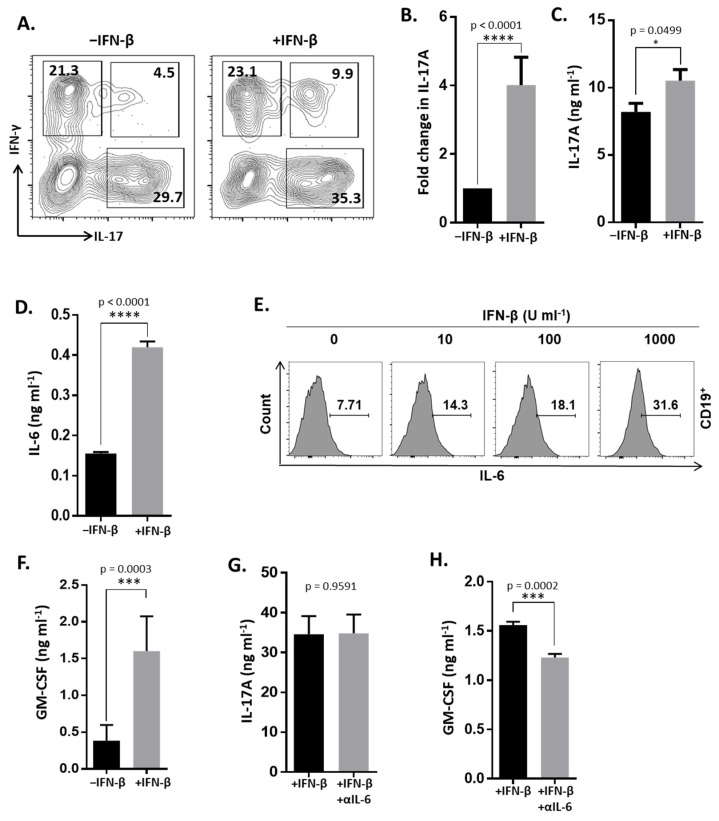
IFN-β induces inflammatory cytokine expression in IL-23-stimulated TH17 cultures. Spleen and lymph node cells of MOG-immunized C57BL/6J mice were cultured with MOG_35–55_, IL-23, and anti-IFN-γ in the presence or absence of IFN-β. (**A**) Representative FACS plots of CD4+ T cells expressing IFN-γ and IL-17. (**B**) Expression of IL-17 transcripts in the cultured cells measured by qRT-PCR. (**C**) Secreted IL-17 levels in the culture supernatants measured by ELISA. (**D**) Secreted levels of IL-6 in the culture supernatants measured by ELISA. (**E**) Intracellular IL-6 was measured in CD19^+^ B cells by flow cytometry. (**F**) GM-CSF in the culture supernatants measured by ELISA. The concentration of (**G**) IL-17 and (**H**) GM-CSF from culture supernatants from IFN-β ± α IL-6 stimulation were assessed by ELISA. Statistical analysis was determined using Mann−Whitney tests. *p* values < 0.05 were considered significant (**** indicates *p* < 0.0001, *** indicates *p* < 0.001, * indicates *p* < 0.05). Error bars indicate s.e.m.

**Figure 3 cells-10-02139-f003:**
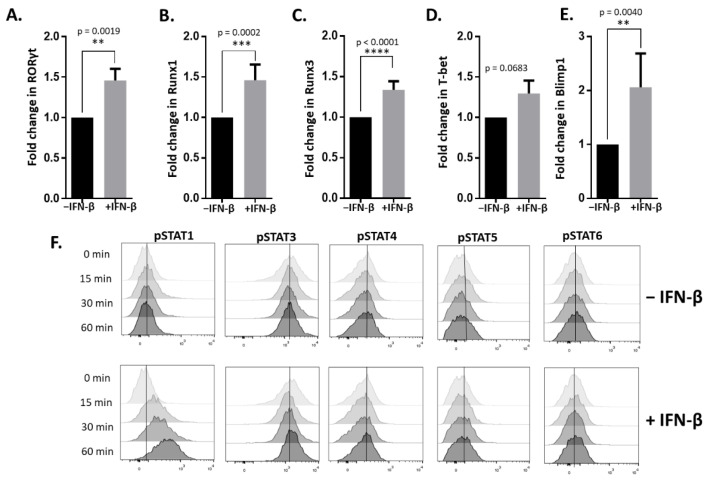
Transcriptional differences in pathogenic IL-23-stimulated TH17 cells induced by IFN-β treatment. Spleen and lymph node cells from MOG_35–55_-immunized C57BL/6J mice were cultured with MOG_35–55_, IL-23 and anti-IFN-γ in the presence or absence of IFN-β for three days. Gene expression was measured by qRT-PCR for the transcription factors; (**A**) RORγt, (**B**) Runx1, (**C**) Runx3, (**D**) T-bet and (**E**) Blimp-1. Cells from the spleen and lymph nodes of MOG-immunized mice were stimulated with MOG_35–55_, anti-IFN-γ, IL-23 ± IFN-β for 0, 15, 30 and 60 min. CD4^+^ T cells were assessed for phosphorylation of (**F**) STAT1, STAT3, STAT4, STAT5 and STAT6 using flow cytometry. The results are representative of two to four independent experiments. Statistical analysis was determined using Mann−Whitney tests. *p* values < 0.05 were considered significant (**** indicates *p* < 0.0001, *** indicates *p* < 0.001, ** indicates *p* < 0.01).

**Figure 4 cells-10-02139-f004:**
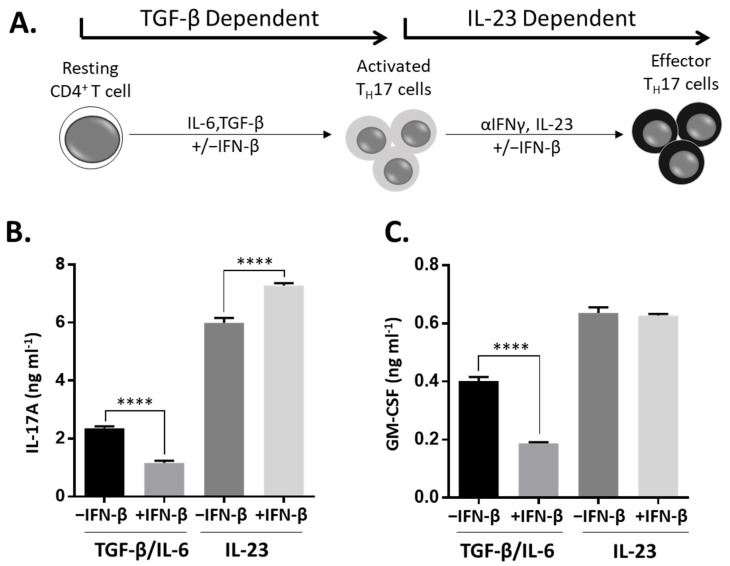
Impact of IFN-β on cytokine production during TGF-β-dependent and IL-23-dependent TH17 differentiation. Spleen and lymph node cells from healthy C57BL/6J were depleted of CD8+ cells and differentiated in the culturing scheme depicted in (**A**). For TGF-β-dependent TH17 differentiation, cells were stimulated with anti-CD3, anti-CD28, IL-6, TGF-β ± IFN-β. For IL-23-dependent TH17 differentiation, cells were stimulated with anti-CD3, anti-CD28, IL-23, anti-IFN-γ ± IFN-β. The concentrations of (**B**) IL-17A and (**C**) GM-CSF secreted were determined by ELISA following early and late differentiation. Statistical analysis was performed using one-way ANOVAs. *p* values < 0.05 were considered significant (**** indicates *p* < 0.0001). Error bars indicate s.e.m.

## Data Availability

All data from this manuscript will be provided to anyone upon request.

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
