# Peer review of "Interferon-β Intensifies Interleukin-23-Driven Pathogenicity of T Helper Cells in Neuroinflammatory Disease"

_cells, 2021, doi:10.3390/cells10082139_

Round 1
Reviewer 1 Report
1) Please, replace "yet" for "however".
2) Do the authors claim that IFN-beta exacerbates neuroinflammation? What about the treatment of patients with RRMS who receive this medication?
Author Response
We thank the reviewer for the constructive comments. They have been addressed below and changes are highlighted in the manuscript.
Comment 1: “Please, replace "yet" for "however".”
- We have changed each “yet” to “however”.
Comment 2: “Do the authors claim that IFN-beta exacerbates neuroinflammation? What about the treatment of patients with RRMS who receive this medication?”
- Our previous publications and work from other research groups have shown that patients with high levels of TH17 signatures, which include a subset of MS patients and NMO patients, are non-responders to IFN-beta therapy and this therapy may increase disease. In these studies, we have also identified that markers in the TH1 pathway are indicative of a good response to IFN-beta therapy in MS.
- This current manuscript provides further mechanisms on how IFN-beta can drive disease activity in NMO and in subsets of IFN-non responder MS patients.
- We have clarified this in the introduction of the manuscript.
Reviewer 2 Report
This is an experimental trial which points out some still unknown MoA of Interferon. Concening the introduction I would recommend that the authors discuss missing compliance of MS patients as further cause for the relatively high rate of patients, who do not respond to interferon therapy in MS. There are data from Germany, who describe this due to the high side effect potential of interferons with flu-like symptoms. The authors may discuss whether this behaviour may contribute to rebound phenomena as a cause for relapses simlar to fingolimod. May this cytokine regulation via interferons may trigger relapses as rebound phenomena in MS ?
Author Response
We thank the reviewer for pointing out that therapeutic compliance may contribute to non-responsiveness to IFN-beta in some patients. We now have mentioned and cite three possibilities for non-response to therapy in the introduction of the manuscript: Non-compliance, neutralizing antibodies, and immunological signatures.
Round 2
Reviewer 2 Report
it is okay now
Author Response
Thank you.